# Metastasis diagnosis using attenuated total reflection-Fourier transform infra-red (ATR-FTIR) spectroscopy

**Samuel Onuh Abuh, Ayan Barbora, Refael Minnes***

Faculty of Natural Sciences, Department of Physics, Ariel University, Ariel, Israel

* refaelm@ariel.ac.il

**Data Availability Statement:** All relevant data are within the manuscript and its Supporting Information files

**Funding:** 'The study was partially funded by a research grant from the Administrator General,

## Abstract

The suitability of Fourier transform infrared spectroscopy as a metastasis prognostic tool has not been reported for some cancer types. Our main aim was to show spectroscopic differences between live un-preprocessed cancer cells of different metastatic levels. Spectra of four cancer cell pairs, including colon cancer (SW480, SW620); human melanoma (WM115, WM266.4); murine melanoma (B16F01, B16F10); and breast cancer (MCF7, MDA-MB-231); each pair having the same genetic background, but different metastatic level were analyzed in the regions 1400–1700 $cm^{-1}$ and 3100–3500 $cm^{-1}$ using Principal Component Analysis, curve fitting, multifractal dimension and receiver operating characteristic (ROC) curves. The results show spectral markers $I_{1540}/I_{1473}$, $I_{1652}/I_{1473}$, $\frac{A_{3400}}{A_{3200}}$, and multifractal dimension of the spectral images are significantly different for the cells based on their metastatic levels. ROC curve analysis showed good diagnostic performance of the spectral markers in separating cells based on metastatic degree, with areas under the ROC curves having 95% confidence interval lower limits greater than 0.5 for most instances. These spectral features can be important in predicting the probability of metastasis in primary tumors, providing useful guidance for treatment planning. Our markers are effective in differentiating metastatic levels without sample fixation or drying and therefore could be compactible for future use in in-vivo procedures involving spectroscopic cancer diagnosis.

## Introduction

Metastasis constitutes the primary cause of death for more than 90% of patients with cancer and a major factor for failure of cancer therapy [1,2]. Most types of primary carcinoma have a five-year survival rate of 80%, but this drops to around 30% once the tumor metastasizes [3]. When choosing a treatment regime for a cancer patient, it is important to correctly identify the tumor's metastatic potential. Currently, diagnosis of metastasis is based mainly on methods such as lymph node status, morphological classification of tumors, imaging, genetic testing of tumors, and molecular analysis of cancer signatures. However, these methods are limited in many cases, such as low specificity and poor negative predictive value of biomarker antigens in prostate and gallbladder cancer [4,5]. Methods based on genetic testing of tumors are

Israel's Ministry of Justice, application number
20220140. The funders had no role in study
design, data collection and analysis, decision to
publish, or preparation of the manuscript. There
was no additional external funding received for this
study.

**Competing interests:** NO authors have competing
interests

expensive and time-consuming [6]. Some tumor markers may be elevated in non-cancerous
conditions, leading to false-positive results [7,8]. Imaging techniques such as CT scans, MRIs,
and PET are expensive, may expose patients to radiation risks [9] and may show low specificity
or poor positive predictive value in some types of cancer [10]. A new method for diagnosis is
therefore desirable.

Fourier transform infrared (FTIR) spectroscopy in the mid-IR (400–4000 $cm^{-1}$) has been
shown to be a suitable diagnostic method for early cancer detection [11–13]. It is highly sensi-
tive, minimally invasive, relatively simple [14], and can give highly reproducible results [15]. It
gives information on the biochemical components (proteins, lipids, carbohydrates, and nucleic
acids) of the material under analysis, detecting metabolic differences between healthy and
cancerous cells and tissues based on variations in these components. However, the presence of
water in biological samples creates challenges for biological applications of mid-IR spectros-
copy. Water's strong absorption within the mid-IR range, especially in the biologically relevant
spectral window, affects signal detection and masks the absorption of cells or tissue compo-
nents [16]. While dry samples can be used to reduce water interference [17], differences exist
between the mid-IR spectra of dehydrated and hydrated cells [18,19]. Fourier Transform Infra-
red (FTIR) spectrometer with an attenuated total reflection (ATR) element helps minimize
this problem while enabling the study of mid-IR absorption of cells and tissues in their
hydrated state [20].

Various studies have employed FTIR spectroscopy to study malignancy in cell lines, includ-
ing melanoma [16,21], colon cancer [22,23] and breast cancer [24–26]. However, many of
these studies employed formalin-fixed and paraffin-embedded (FFPE) cells. It has been shown
that paraffin can interfere with FTIR measurements due to its strong absorption of infrared
radiation. Also, deparaffinization procedures typically performed on samples can result in the
loss of spectral information about free and unbound lipids as well as the potential modification
of protein contents [27–29]. FTIR measurements are therefore best carried out on fresh tissues
or live cells in their unprocessed state.

In previous works from our laboratory [16,30], significant ATR-FTIR spectroscopic differ-
ences between cancer cells based on their metastatic levels were presented. Membrane hydra-
tion and amide I protein intensity were employed to show difference in metastatic potential in
human and in murine melanoma [16] while multifractal dimension of spectral images was
used to differentiate murine melanoma cell pair; and colon cancer cell pair [30] of different
metastatic potentials. The markers based on membrane hydration level have not been investi-
gated for colon and breast cancer cells, while that of the spectral multifractal dimension has
not been investigated for human melanoma and breast cancer cells. Here, the spectroscopic
markers of metastasis presented in these prior works have been tested for their discriminatory
performance in each of the cell lines including colon cancer, human melanoma, murine mela-
noma, and a new pair, breast cancer cell lines. The previous work results showed that higher
metastatic potential correlates with membrane hydration level, protein absorption intensity
and spectral multifractal dimension. Other groups have employed water absorption related
signatures to study protein hydration and to distinguish oral cavity squamous cell carcinoma
from neighboring healthy tissues [31–33]. Using ATR-FTIR spectroscopy, Shinzawa's group
[31] showed the existence within mammalian cells of distinct water absorption peaks around
3420 and 3220 $cm^{-1}$ and that these peaks varied spatially, depending on interaction with intra-
cellular proteins. In a separate study, increased Raman bands of the OH-stretching vibrations
was observed in squamous cell carcinoma compared to surrounding healthy tissues [32]. This
difference in water concentration was utilized to differentiate between tumor and surgical
margins of oral cavity squamous cell carcinoma using the ratio of the Raman bands at 3390
and 2935 $cm^{-1}$ [33]. Spectroscopic differences between cells of different metastatic levels can

serve to complement current methods of metastasis diagnosis as well as minimize limitations associated with them. In this study, we aim to employ univariate and multivariate analysis of proteins and water absorption related FTIR spectroscopic biomarkers to evaluate differences between colon cancer, human and murine melanoma, and breast cancer cell lines of different metastatic level.

## Materials and methods

### Cell lines, culture and preparation

WM-115 and WM-266.4 cells are human melanoma cells isolated from the primary tumor and a metastasis of the same patient. WM-266.4 cells are known for their aggressive subcutaneous growth in mice [34]. SW480 and SW620 are, respectively, primary tumor and metastasis derived human colon carcinoma cells isolated from a single patient. B16F01 and B16F10 are murine melanoma cells of the same parental origin. B16F10 are known to be more metastatic than B16F01 [16]. MCF7 and MDA-MB-231 are non-metastatic malignant and metastatic breast cancer cells, respectively.

For all the cell lines used, growth protocol and medium information is as presented in [16]. The cells were then washed 1x with PBS (02-023-1A Biological Industries, Beit Haemek, Israel) and then removed with Trypsin EDTA solution B (03-052- 1B Biological Industries, Beit Haemek, Israel). The cells were spun down and re-suspended in normal growth medium in final concentrations of $1 \times 10^5$, $2 \times 10^5$, $4 \times 10^5$, and $1 \times 10^6$ cells per ml for all the cell lines and a single concentration of $1 \times 10^6$ cells per ml for the human melanoma cells.

### ATR-FTIR spectral measurement

FTIR measurements were carried out using an FTIR spectrometer (Jasco, FTIR 6800) equipped with a single-reflection diamond ATR device with an effective dimensions of 1.8 mm diameter. The refractive index of the diamond is 2.42, and the angle of incidence is 45 degrees, generating a single reflection. At 1000 cm$^{-1}$, the calculated depth of penetration is 2.005 μm, assuming a sample refractive index of 1.5. The instrument is coupled to a liquid nitrogen-cooled MCT (mercury cadmium telluride) detector. Measurements were carried out in the spectral range 650–4000 cm$^{-1}$ and with a resolution of 4 cm$^{-1}$, coadding 64 scans for every spectrum.

For each cell concentration level, three spectra measurements were obtained while for the human melanoma, each measurement was recorded in five spectra replicates. A medium layer of 10μl was placed on the diamond ATR crystal after background measurement. The background measurement was obtained using a cell-free suspension medium. Prior to every new sample measurement, the ATR crystal was cleaned with 70% ethanol, and a new background measurement was taken.

### Data processing

Obtained spectra were cut to the region 1400–1700 cm$^{-1}$ wavenumbers, followed by simple baseline subtraction and vector normalization. Calculations and data processing procedures were performed using OriginPro software (OriginPro Version 2023b, OriginLab Corporation, Northampton, MA, USA). Curve fitting of a Gaussians-sum model to the spectra was performed using the non-linear curve fitting algorithm implemented in OriginPro software. The FracLac plugin (version2015) of the ImageJ software version 1.54f (Image Processing and Analysis in Java–Wayne Rasband and contributors, National Institutes of Health, Bethesda,

MD, USA, public domain license, https://imagej.nih.gov/ij/) was used to perform multifractal analysis of the preprocessed spectral images of all samples.

The ROC curve analysis was carried out using the ROC curve algorithm in OriginPro. Given that $x_-$ denotes x values for cases with negative actual states and $x_+$ for cases with positive actual states (where x is the scale of the test result variable), the AUC is calculated as the non-parametric approximation of the area under the ROC curve A, such that:

$$A = \frac{1}{n_+ n_-} \sum_{j=1}^{n_-} \sum_{i=1}^{n_+} \Psi\left(x_+ x_-\right) \tag{1}$$

Where $n_+$ is the sample size of positive instances (more metastatic), $n_-$ is the sample size of negative instances (less metastatic) and

$$\Psi\left(x_+, x_-\right) = \begin{cases} 1, & \text{if } x_+ > x_- \\ 0.5, & \text{if } x_+ = x_- \\ 0, & \text{if } x_+ < x_- \end{cases} \tag{2}$$

The cutoffs used to construct the ROC curves were determined by the interpolation of data points method and defined as the midpoints between successive pairs of ordered marker scores $x_i$, with two additional points, $\max(x_i) + 1$ and $\min(x_i) - 1$, where $\max(x_i)$ and $\min(x_i)$ represents the maximum and minimum scores respectively for a given marker.

## Results

### The absorption intensities of proteins increase with metastatic level of cancer cells

Fig 1 presents the mean preprocessed absorbance spectra in the region 1400–1700 cm$^{-1}$ for all cell lines used in this study. Preprocessing steps included simple baseline subtraction and vector normalization. Vector normalization was used to allow for amide II peak comparison of the cells since the amide II band was observed to be the most intense peak [35].

The peaks at 1540 and 1652 cm$^{-1}$ are associated with the Amide II and Amide I proteins, respectively, while the peak at 1456 cm$^{-1}$ is related to the asymmetric CH$_3$ bending vibrations of the methyl groups of proteins [36,37]. Table 1 presents the relative intensities ($I_{peak1}/I_{peak2}$) of these protein-related peaks with respect to a peak around 1473 cm$^{-1}$, related to the CH$_2$ bending of the methylene chains in lipids [38].

The measured intensity ratios $I_{1456}/I_{1473}$, $I_{1540}/I_{1473}$, and $I_{1652}/I_{1473}$ are shown in a bar chart in Fig 2 for all the cell pairs. Bars represent mean values and error bars represent standard errors of the means for this figure and all charts presented in this study. For all cell pairs investigated, the ratios were higher for the more metastatic cells than their corresponding less metastatic cells, except for the breast cancer cell pair. Higher relative protein intensities observed for metastatic colon, human melanoma, and murine melanoma cells are consistent with observations reported by [16] for murine melanoma (B16-F1 and B16-F10) cells and human melanoma cells (WM-115 and WM-266.4). The study showed higher amide II intensities for more metastatic cells compared to less metastatic ones and suggested the difference in protein intensities could indicate the membrane hydration level (and hence motility) of the cells. The protein peaks for the breast cancer cells showed higher intensities in the malignant MCF7 cells than in the metastatic MDA-MB-231 cells. A similar observation was reported by [25] when they compared the FTIR spectra of three types of breast cancer cells, including the MCF7 and the MDA-MB-231. This could be due to the fact that MDA-MB-231 is a triple negative breast cancer cell line and lacks certain receptors, which may affect their protein expression profiles.

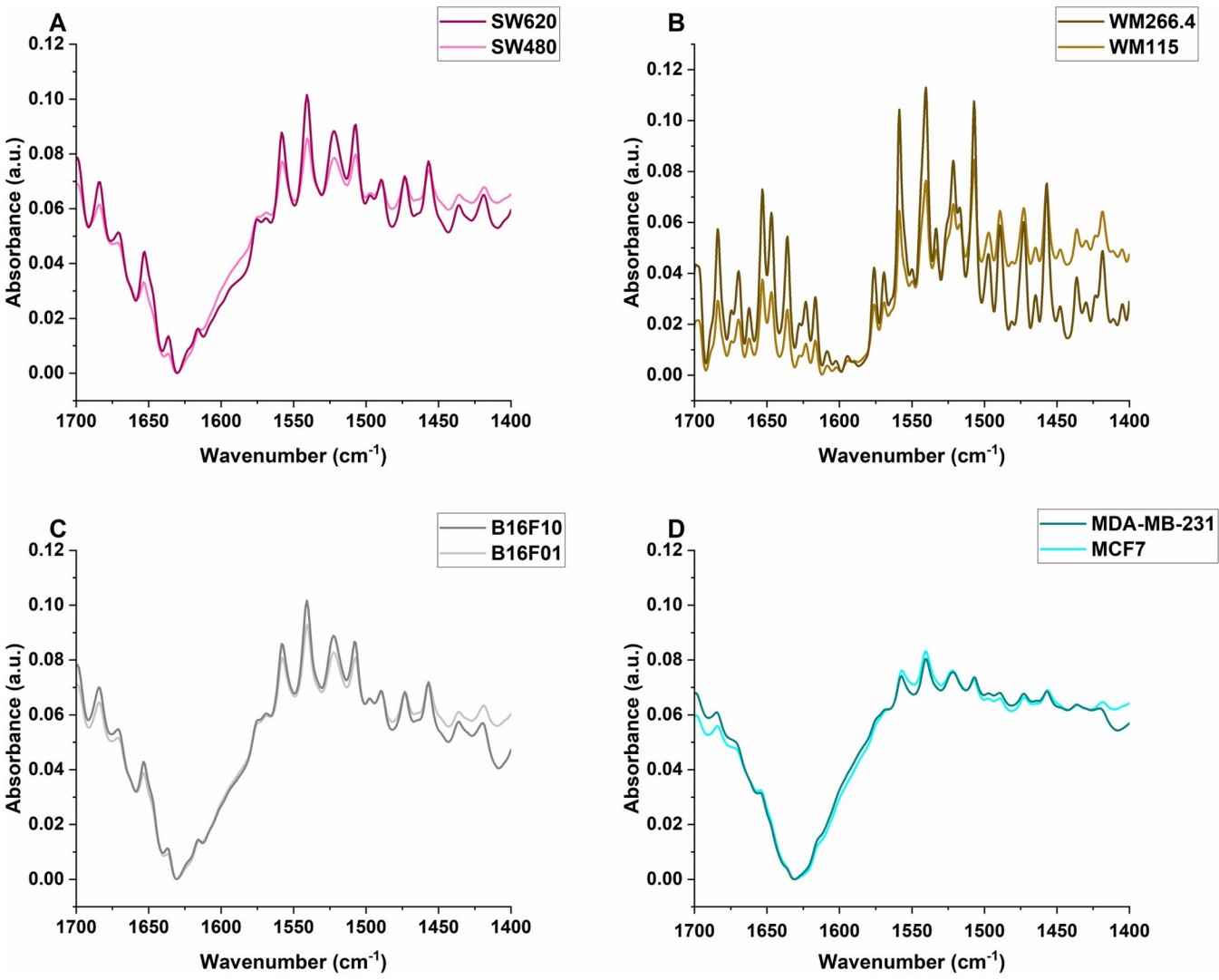

**Fig 1. Mean ATR-FTIR absorption spectra for all cell lines in the region 1400–1700 cm$^{-1}$.** (A) SW480 and SW620. (B) WM115 and WM266.4 (C) B16F01 and B16F10 (D) MCF7 and MDA-MB-231.

## PCA for the 1400–1700 cm$^{-1}$ region separates cancer cells based on metastatic level

A PCA was applied to the spectral data in the 1400–1700 cm$^{-1}$ region for all cell pairs. Fig 3 presents scatterplots of the percentage of each sample spectrum captured along the first two principal component axes, with each point on the plot representing a sample. This approach presents the scores with a more intuitive interpretation than the usual positive and negative scores.

To obtain these percentages, the scores obtained from PCA were first transformed along the PC axes by adding the respective score ranges across samples in order to remove negative values. Given that the spectra data X, can be written in terms of the scores and principal components as:

$$X = \alpha_1 PC_1 + \alpha_2 PC_2 + \alpha_3 PC_3 + \alpha_k PC_k + \cdots + \alpha_n PC_n \tag{3}$$

**Table 1. Relative intensities of the peaks at 1456, 1540, and 1652 $cm^{-1}$ with respect to the intensity of the peak at 1473 $cm^{-1}$.**

|  | $I_{1456}/I_{1473}$ | $I_{1540}/I_{1473}$ | $I_{1652}/I_{1473}$ |
|---|---|---|---|
| SW480 | 1.045 ± 0.007 | 1.21 ± 0.03 | 0.46 ± 0.02 |
| SW620 | 1.076 ± 0.002 | 1.412 ± 0.008 | 0.616 ± 0.005 |
| p-value* | 0.0002 | <0.0001 | <0.0001 |
| WM115 | 1.13 ± 0.02 | 1.2 ± 0.1 | 0.57 ± 0.08 |
| WM266.4 | 1.26 ± 0.03 | 1.9 ± 0.2 | 1.2 ± 0.2 |
| p-value* | <0.0001 | <0.0001 | <0.0001 |
| B16F01 | 1.053 ± 0.003 | 1.36 ± 0.03 | 0.56 ± 0.02 |
| B16F10 | 1.052 ± 0.006 | 1.49 ± 0.03 | 0.62 ± 0.02 |
| p-value* | 0.507 | 0.007 | 0.026 |
| MCF7 | 1.040 ± 0.005 | 1.26 ± 0.03 | 0.489 ± 0.007 |
| MDA-MB-231 | 1.015 ± 0.002 | 1.19 ± 0.01 | 0.461 ± 0.007 |
| p-value* | 0.001 | 0.260 | 0.157 |

*Significant if the p-value is less than 0.05 (p<0.05) using the Mann-Whitney U test.

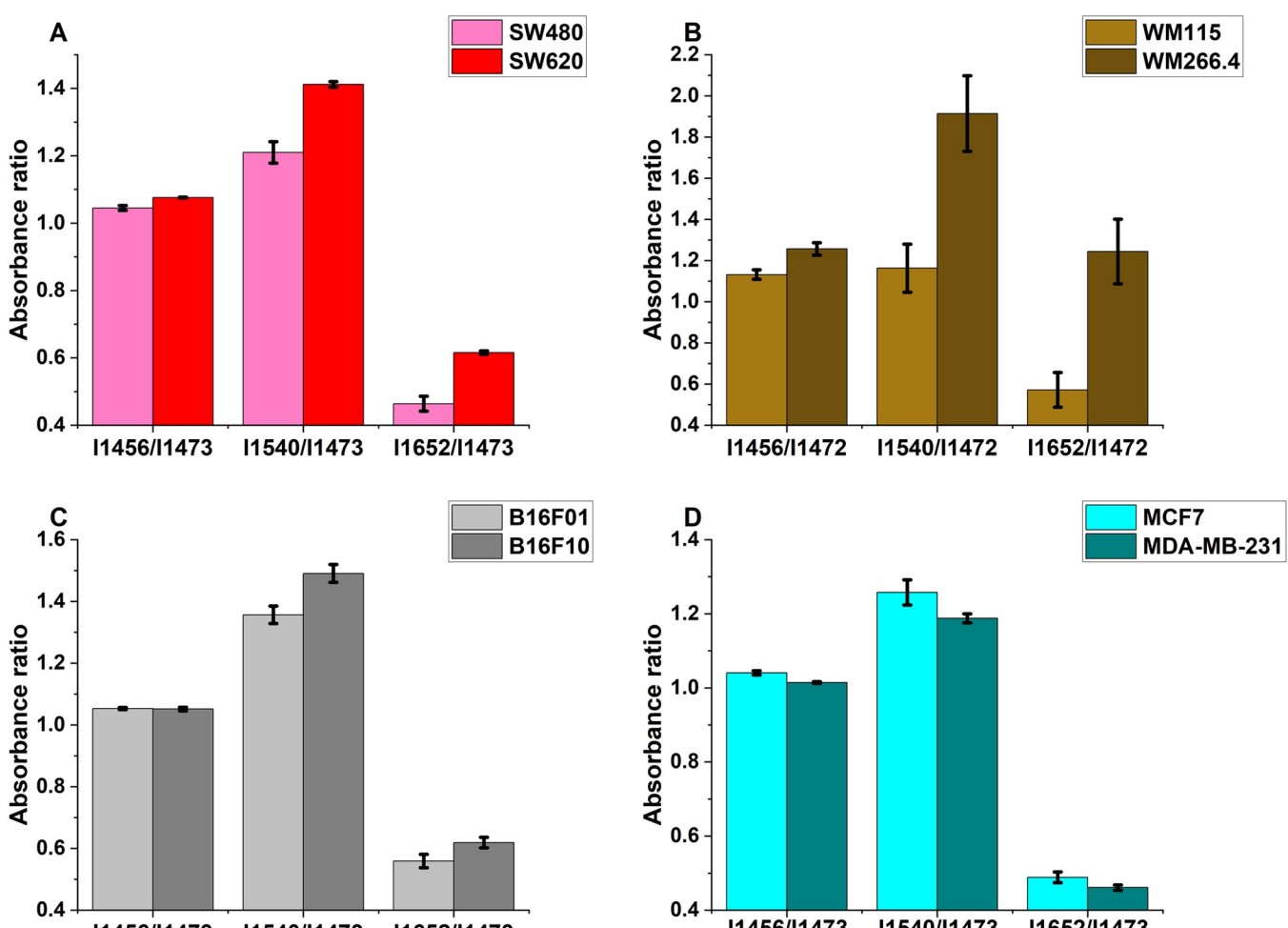

**Fig 2. Relative intensities of the peaks at 1456, 1540, and 1652 $cm^{-1}$ with respect to the intensity of the peak at 1473 $cm^{-1}$.** (A) SW480 and SW620 (B) WM115 and WM266.4 (C) B16F01 and B16F10 (D) MCF7 and MDA-MB-231 cells.

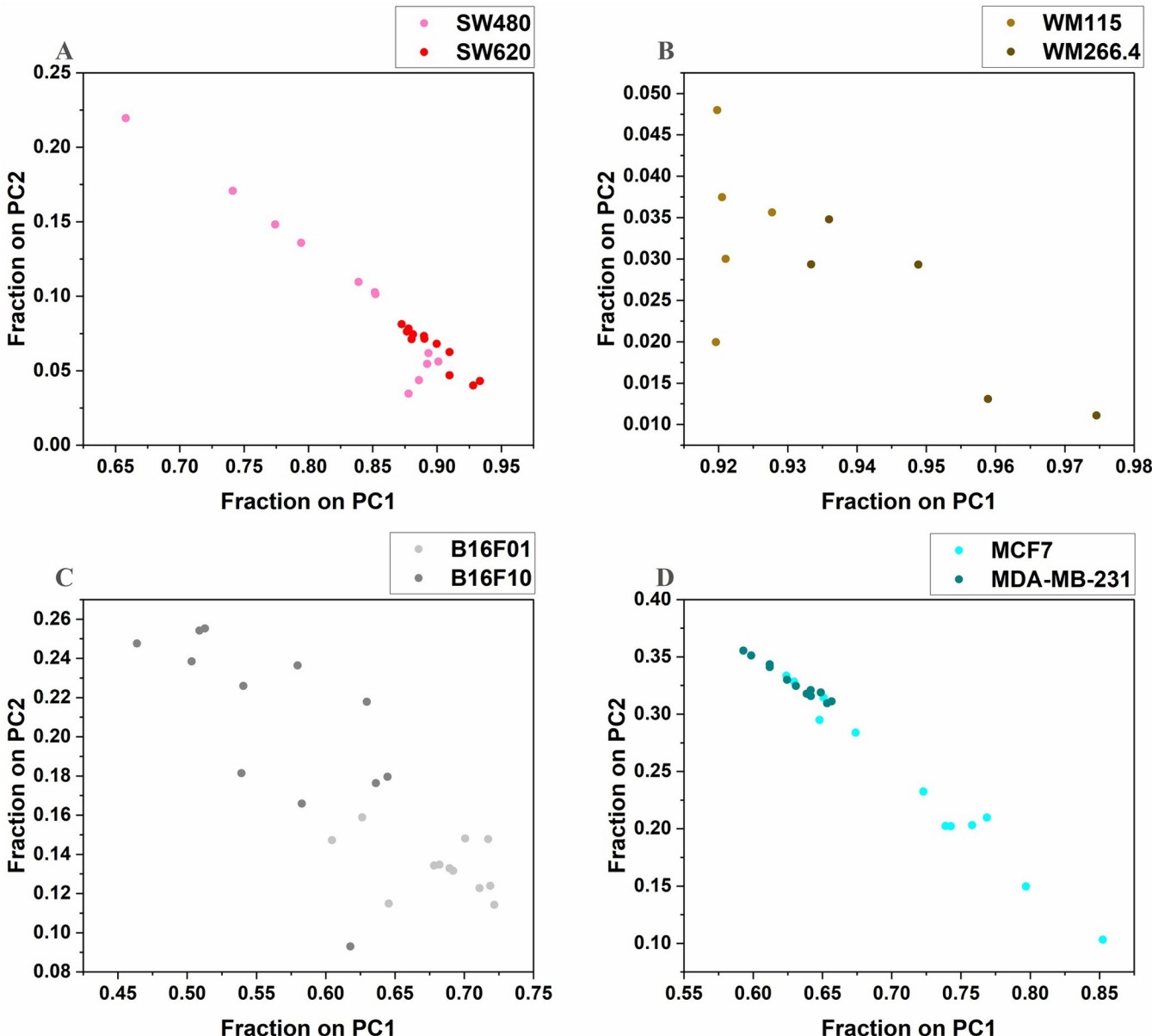

**Fig 3. A scatter plot of percentage of sample spectra captured on PC1 and PC2.** (A) SW480 and SW620 cells (B) WM115 and WM266.4 (C) B16F01 and B16F10 (D) MCF7 and MDA-MB-231.

Where $\alpha_k$ represents the k[th] score/projection of a given sample onto the k[th] *PC*. The percentage of each sample accounted for along a given PC was computed as:

$$fraction\ of\ X\ on\ PC_k = \frac{\alpha_k/\boldsymbol{PC}_k/}{\sum_1^n (\alpha_n/\boldsymbol{PC}_n/)} \tag{4}$$

Where $/PC_k/$ represents the Euclidian norm of the k[th] loading vector, $PC_k$, and n is the number of PCs for which 100 percent of variability within the data is accounted for.

The more metastatic cells showed higher percentage values along PC1 (Fig 3). This indicates that these cells contain relatively higher levels of the functional groups which PC1

represents, including the protein peaks at 1540 and 1652 cm$^{-1}$. Fig 3C shows more metastatic B16F10 clustering towards lower percentages than the less metastatic B16F01 along the PC1 axis. However, its loading vector (Fig E in S1 File) pointed in the negative direction, and one can conclude that the lower percentages correspond to higher contents of the functional groups represented by the PC1 axis, including the amide I and II proteins. MCF7 and MDA-MB-231 cells (Fig 3D) similarly presented negative loadings (Fig G in S1 File) indicating the malignant MCF7 cells have relatively higher amide I and II protein content than the metastatic MDA-MB-231 cells.

## The intermolecular structure of water distinguishes cells based on metastatic levels

Water molecules can form two types of structures: low-density water (LDW), with water hydrogen-bonded to other water molecules, forming crystal-like patterns; and high-density water (HDW), with water bonded to non-water molecules. The different bonding patterns give rise to variations in water IR vibrations, particularly in the O-H stretch mode. This difference in vibration leads to distinct absorption peaks for LDW and HDW. It has been shown that liquid water can be thought of as being composed of a mixture of these structural types [39]. Each type has its characteristic O-H stretch mode absorption peak; the LDW is around 3200 cm$^{-1}$ and the HDW is around 3400 cm$^{-1}$. The HDW-to-LDW ratio can measure the difference in cell membrane hydration level. This is because a higher membrane hydration level (i.e., water molecules are in interaction with a hydrophilic surface) implies a higher probability of breaking the intermolecular hydrogen bonds that form the structural order [16,40].

To access the HDW-to-LDW ratio, an analysis based on a two-component mixture model presented by Minnes et al. [16] (a simplified version of the 4-component mixture model from [41] was adopted. The absorption spectrum of each measurement in the range of 3100–3500 cm$^{-1}$ was fitted to a mathematical model consisting of the sum of two Gaussians using the Levenberg–Marquardt iterative optimization algorithm. The model is of the form $y_o +$

$\frac{A_1}{w\sqrt{\frac{\pi}{2}}}e^{-2\frac{(x-\mu_1)^2}{w^2}} + \frac{A_2}{w\sqrt{\frac{\pi}{2}}}e^{-2\frac{(x-\mu_2)^2}{w^2}}$ (the area version of the gaussian function in Origin software), where $y_0$ = offset, $\mu_1$ and $\mu_2$ = peak centers (set to be around 3200 and 3400 cm$^{-1}$, respectively), w = peak width = $2\sigma$, and A = area. An example of the absorption spectrum and its fitted Gaussians is shown in Fig 4, with an adjusted R$^2$ value of 0.9976.

The ratio of the area of the Gaussian centered around 3400 cm$^{-1}$ to that of the Gaussian centered around 3200 cm$^{-1}$ was obtained as $\frac{A_{3400}}{A_{3200}}$. The average area ratio for SW480 was found to be $6.7 \pm 0.1$ and for SW620 it was $5.9 \pm 0.2$. The average area ratio for WM115 was found to be $18.3 \pm 0.7$ and for WM266.4 it was $12.5 \pm 1.4$. $\frac{A_{3400}}{A_{3200}}$ showed averages of $6.3 \pm 0.2$ and $5.8 \pm 0.2$ for the B16F01 and B16F10 cells; $7.4 \pm 0.2$ and $7.0 \pm 0.2$ for MCF7 and MDA-MB-231 cells, respectively. The Mann-Whitney U test indicated the ratios do significantly tend to be less for the more metastatic than the less metastatic colon and human melanoma cells (p values at 0.05 significance level were 0.003, 0.002, 0.253, and 0.130 respectively, for colon, human melanoma, murine melanoma, and the breast cancer cells). Using the proposal by [16], for negative water peaks such as those observed in this experiment, we can assume that $\frac{A_{3400}}{A_{3200}} \propto \left(\frac{N_{HDW}}{N_{LDW}}\right)^{-1}$. Therefore, the area ratios show that the less metastatic cells have more LDW (or less HDW) than the more metastatic ones, as previously reported in [16]. This suggests that for the more metastatic cells, in comparison with the less metastatic, more water molecules are in interaction with the hydrophilic cell membrane surface. A bar chart of the water area ratios is presented in Fig 5.

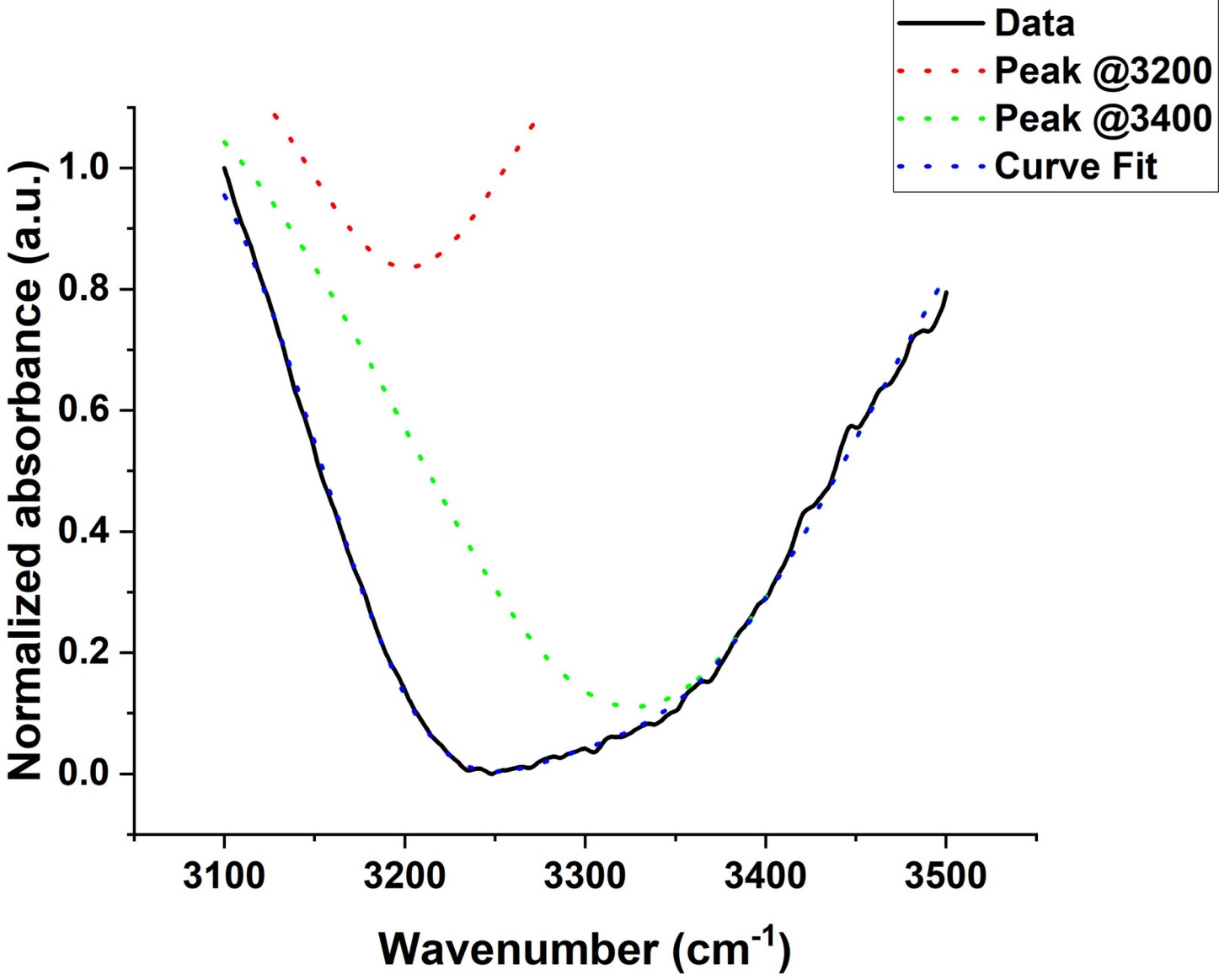

**Fig 4. An example of the two-Gaussian fit model for one of the samples.**

### Multifractal dimension of spectral image increases with metastatic level

The multifractal dimension f($\alpha$) was obtained from the multifractal spectrum curve f($\alpha$) vs $\alpha$, at the limit Q = 0, f($\alpha$) = capacity dimension = box counting dimension, where Q represents a range of exponents used to calculate the variables in multifractal analysis, and $\alpha$, the singularity exponent. It has been shown that the multifractal dimensions of ATR-FTIR spectrum can identify and quantify cancer cell metastatic level [30]. The average fractal dimensions for each cell pair are presented in Fig 6, with the spectral images of the more metastatic cells showing higher fractal dimensions compared to those of less metastatic cells (p values at 0.05 significance level were 0.001, <0.001, 0.03, and 0.19 for colon, human melanoma, murine melanoma, and breast cancer cells respectively, using the Mann-Whitney U test). This shows that the IR

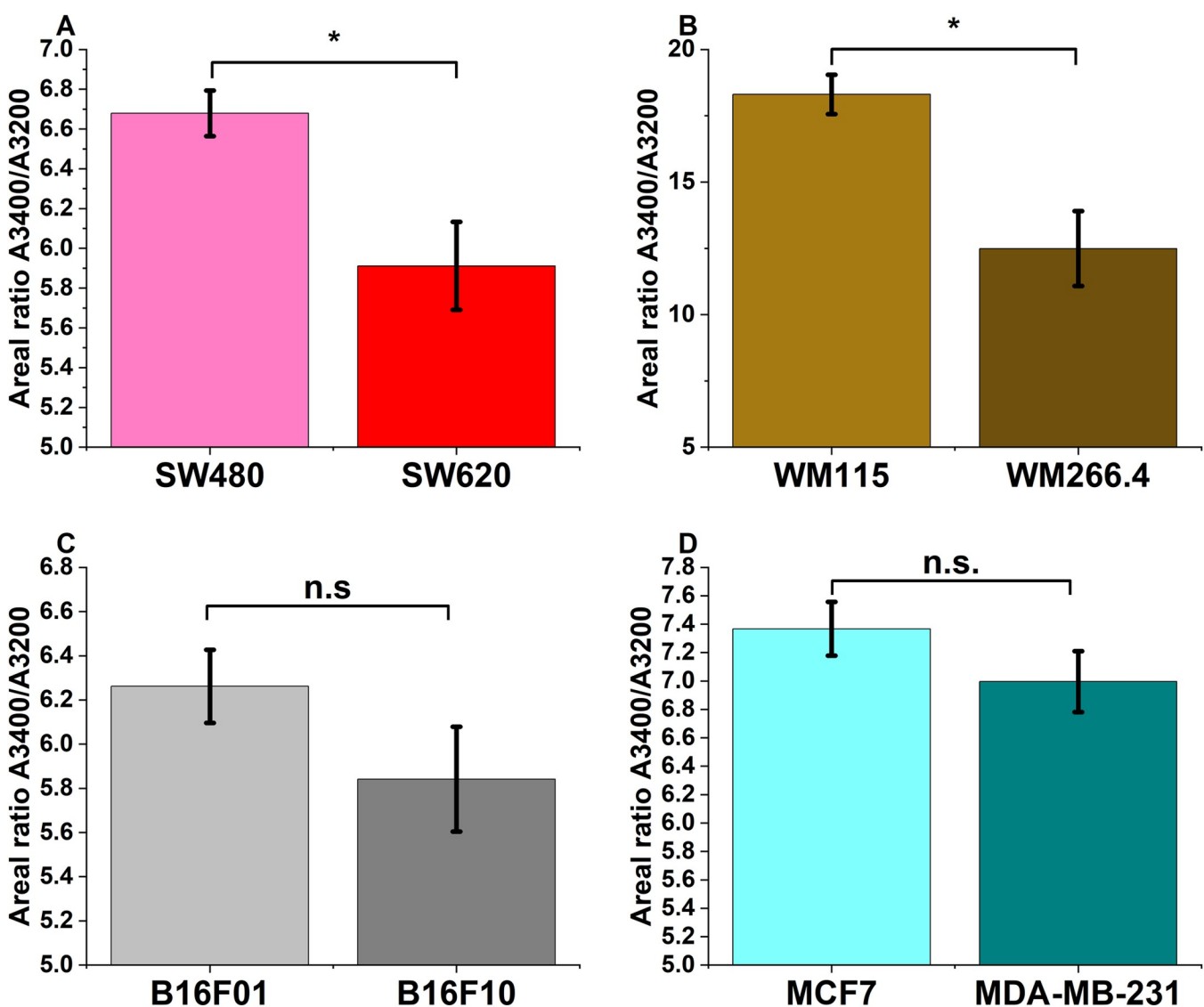

**Fig 5. Mean area ratios of the fitted Gaussians centered around 3400 and 3200 cm⁻¹.** (A) SW480 and SW620 cells (B) WM115 and WM266.4 (C) B16F01 and B16F10 (D) MCF7 and MDA-MB-231. * represents statistically significant differences at p<0.05 using the Mann-Whitney U test, while n.s. shows lack of statistical significance at the same p threshold.

absorption profiles of less metastatic and more metastatic cells show varying degrees of self-similarity that may be due to the variations in peak heights, positions, or broadness.

## Receiver operating characteristic (ROC) curve analysis of spectral markers of metastasis

The discrimination between less metastatic and more metastatic cells was further evaluated using ROC curve analysis. Fig 7 presents the performance of each marker for colon cancer, human melanoma, murine melanoma, and the breast cancer cell lines. The accuracy of each of the spectroscopic marker including the relative peak at 1540 cm⁻¹, water areal ratio $\frac{A_{3400}}{A_{3200}}$ and the multifractal dimension of the spectral images, was determined using the area under the ROC curve (AUC-ROC). The sensitivity and specificity for the detection of the more

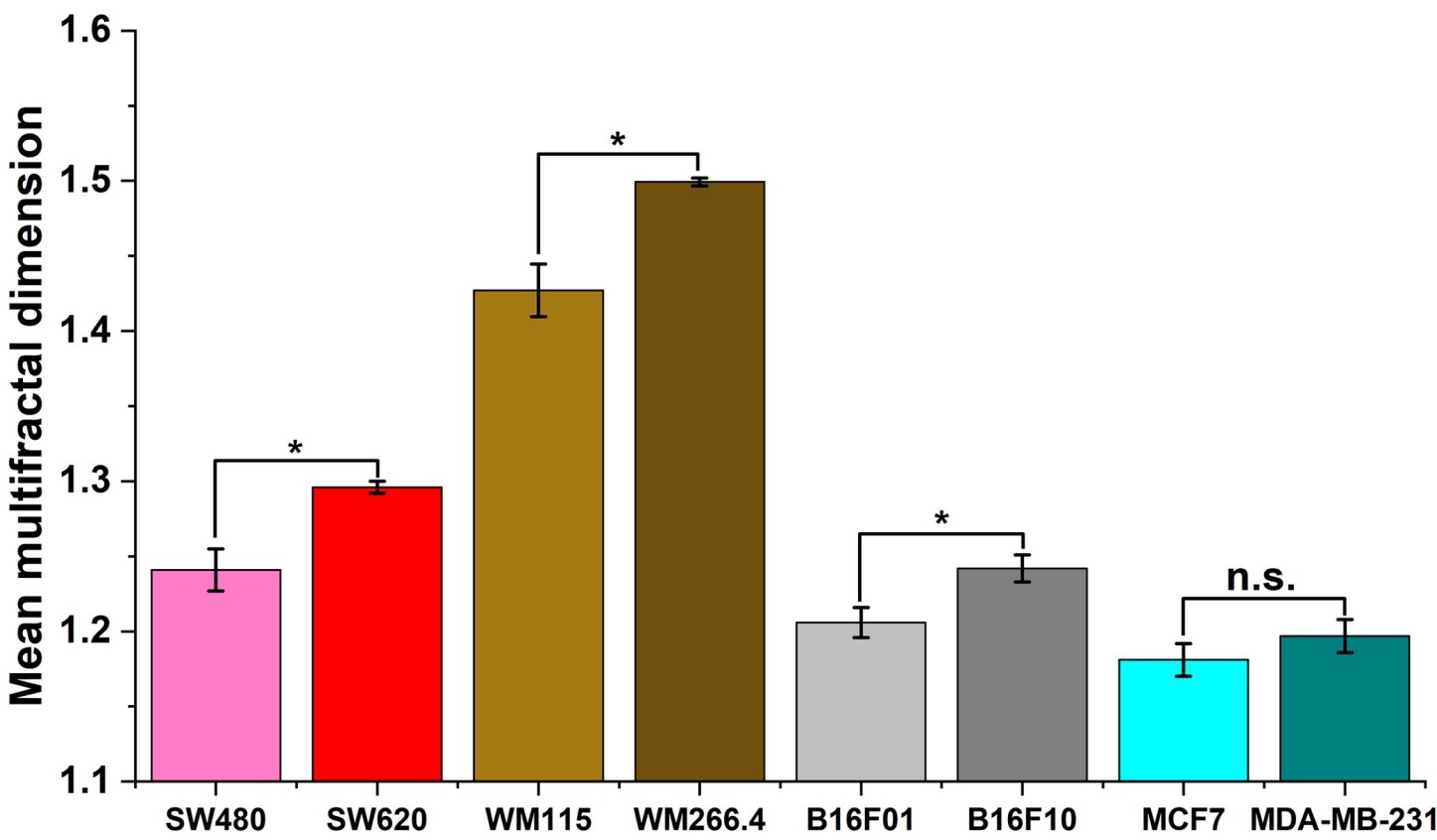

**Fig 6. Mean fractal dimensions of images of sample spectra in the region 1400–1700 cm$^{-1}$ for more metastatic and less metastatic cell pairs.** * represents statistically significant differences at p<0.05 using the Mann-Whitney U test, while n.s. shows lack of statistical significance at the same p threshold.

metastatic cells relative to the less metastatic ones are presented in Table 2, along with the area under the curve (AUC).

Fig 7 shows plots of sensitivity (the true positive rate) against the false positive rate (1 - specificity) at different thresholds. Point (0,0) represents 0% sensitivity and 100% specificity; (1,0) represents 100% sensitivity, 100% specificity; (1,1) represents 100% sensitivity and 0% specificity while (0,1) represents 0% sensitivity, 0% specificity. The top-left corner (1,0) therefore represents the best point corresponding to a perfect classifier that correctly identifies all positive cases while avoiding all false positives.

The AUC-ROC is a scalar value which estimates the overall model accuracy, independent of threshold. It is probability that the classifier will rank a randomly chosen positive instance higher than a randomly chosen negative instance or the average value of sensitivity for all possible values of specificity. It ranges from 0 to 1, indicating poor to perfect performance respectively while 0.5 indicates performance equivalent to random chance. The reference line segment from (0,0) to (1,1) called the chance diagonal has an AUC of 0.5 and represents the practical lower limit for the AUC-ROC of an acceptable diagnostic test. Table 2 shows AUC values indicating good marker accuracy within an acceptable 95% confidence interval for most of our markers.

To further evaluate the classification performance of the presented markers, the transformed scores on the first four principal components according to Eq 4, along with the ratios $I_{1540}/I_{1473}$, $I_{1652}/I_{1473}$, $\frac{A_{3400}}{A_{3200}}$ and the multifractal dimensions were used as input for a linear

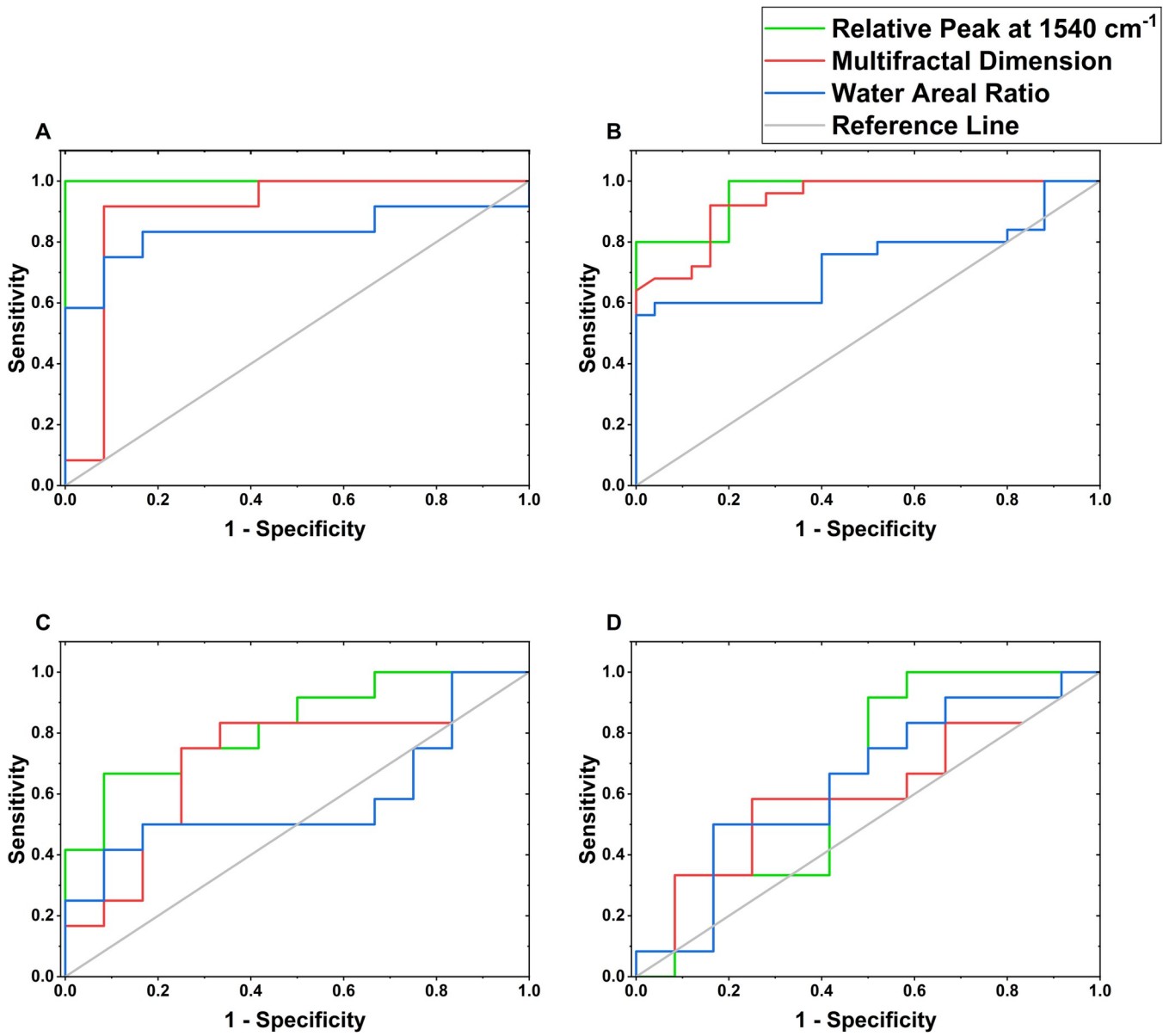

**Fig 7. Receiver operating characteristic curves (ROC) showing diagnostic accuracy of spectral markers.** (A) Colon cancer (B) Human melanoma (C) Murine melanoma (D) Breast cancer.

discriminant analysis (LDA) model. Using the leave one out cross validation, an accuracy of 95.8%, with sensitivity and specificity of 100.0% and 91.7% respectively, were obtained for the colon cancer cells. For the murine melanoma, accuracy, sensitivity, and specificity respectively, were 83.3%, 75.0%, and 91.7%. For the breast cancer cells, 70.8% accuracy, 66.7% sensitivity, and 75.0% specificity were obtained. The number of variables was more than the sample size for the human melanoma, and it was left out of the classification analysis. A plot of the scores on the canonical variable axis is presented in Fig 8 for the analyzed cell types, with the more metastatic well separated from the less metastatic cells.

**Table 2. Sensitivity, Specificity, and Accuracy (AUC) of each spectroscopic marker for the different cancer cells.**

| | | Relative Peak at 1540 cm$^{-1}$ | Water Areal Ratio | Fractal Dimension |
|---|---|---|---|---|
| **Colon Cancer** | **Sensitivity** | 1.00 | 0.92 | 0.92 |
| | **Specificity** | 1.00 | 0.83 | 0.92 |
| | **AUC (95% CI)** | 1.00 (1.00 to 1.00) | 0.83 (0.64 to 1.02) | 0.90 (0.74 to 1.05) |
| **Human Melanoma** | **Sensitivity** | 1.00 | 0.76 | 1.00 |
| | **Specificity** | 0.80 | 0.56 | 0.64 |
| | **AUC (95% CI)** | 0.96 (0.84 to 1.08) | 0.74 (0.59 to 0.89) | 0.93 (0.87 to 0.99) |
| **Murine Melanoma** | **Sensitivity** | 0.92 | 0.58 | 0.83 |
| | **Specificity** | 0.50 | 0.33 | 0.67 |
| | **AUC (95% CI)** | 0.83 (0.66 to 0.99) | 0.58 (0.34 to 0.83) | 0.72 (0.51 to 0.94) |
| **Breast Cancer** | **Sensitivity** | 0.92 | 0.75 | 0.67 |
| | **Specificity** | 0.50 | 0.50 | 0.42 |
| | **AUC (95% CI)** | 0.64 (0.40 to 0.88) | 0.64 (0.41 to 0.87) | 0.61 (0.37 to 0.84) |

## Discussion

Here, we extended the studies in [16,30] by investigating the discrimination performance of the therein proposed ATR-FTIR spectral markers of metastasis, for distinguishing melanoma, colon cancer and for the first time, breast cancer cells of different metastatic levels. Our new results show good classification performance based on the tested markers, suggesting that these markers may be important spectral indicators for the prognosis of metastasis. The best classification performance of the markers was for the colon cancer cells with accuracy of 95.8%. Colorectal cancer remains a high mortality malignancy for which clinicopathological prognostic indicators like lymphovascular invasion are insufficient especially for stage II disease, while multigene expression indicators are limited by high cost and complex scoring threshold system [42]. Spectroscopic markers such as those analyzed here can be useful in providing cheap and rapid complementary indicators to improve the prognosis of colon cancer and ultimately patient outcome. AUC-ROC and LDA classification also show promising accuracy for the two melanoma cell types. The clinical utility of gene expression profile for the identification of stage I melanoma patients at risk of recurrence remains unclear [43] while pathological prognosis can be challenging and may be subjective especially in terms of measuring features like tumor thickness [44]. The biochemical contrast provided by our markers can therefore serve as an additional tool for the pathologist for staging and prognostic purposes. The breast cancer cell lines showed lower confidence interval of AUC-ROC of the investigated markers that were generally below 50 percent (Table 2). Although combining the markers by LDA improved the accuracy for discriminating the breast cancer cells, the sensitivity towards the more metastatic breast cancer cell line was low at 66.7%. This may imply the markers here analyzed may not be universal and could be less useful for monitoring metastasis in breast cancer.

The presented spectroscopic differences in metastasis levels can be useful in predicting the probability of metastasis in primary tumors, for rapidly identifying primary tumors that are at premetastatic stage, providing important guidance for therapy and treatment planning. The discriminating spectral features ($I_{1540}/I_{1473}$, $I_{1652}/I_{1473}$, and $\frac{A_{3400}}{A_{3200}}$) provide complementary evidence that the membrane hydration level is higher for cells of higher level of metastasis as earlier reported in [16]. The amide I and II peaks may therefore be considered significant measures of the spectroscopic difference between cells at different levels of metastasis. The

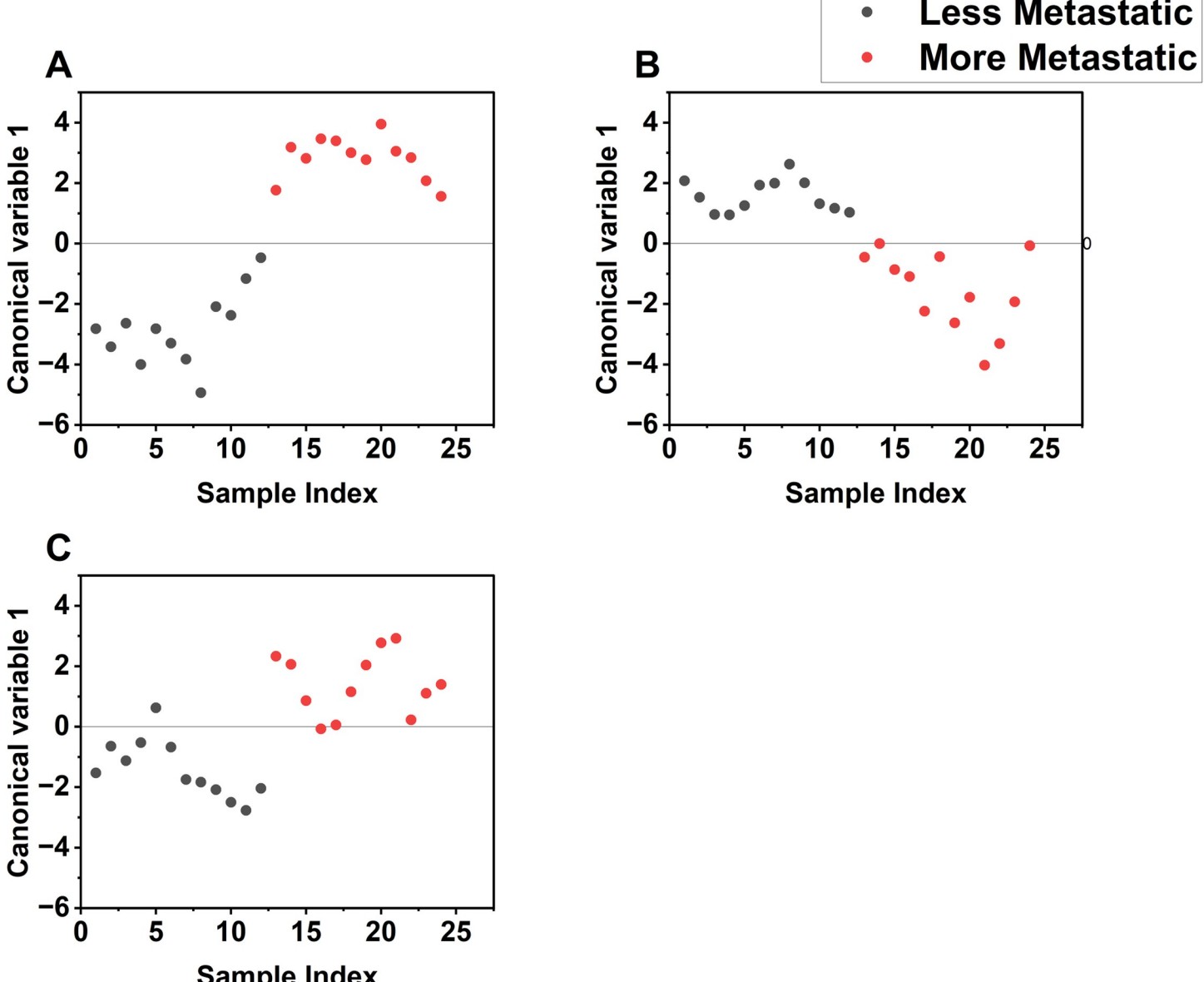

**Fig 8. LDA score plots showing separation of cells according to metastatic level.** (A) Colon cancer (B) Murine melanoma (C) Breast cancer.

relative intensity $I_{1456}/I_{1473}$ did not show consistency in discriminating the cell pairs at the requisite significance level as can be seen in its values for the B16 cells in Table 1. This was further indicated by its low loading coefficients along the first principal components in PCA. This ratio may therefore not be very useful in differentiating cells based on their level of metastasis.

The experiments were performed using live cells without the complications of fixation or freezing. This is important as most sample preparation procedures may not be compatible for any future in-vivo diagnostic or therapeutic procedures involving spectroscopic diagnosis of primary cancer or delineation of cancer margins during surgeries. A spectroscopic technique that will be useful for these procedures must be able to discriminate disease states in the presence of barriers such as tissue wetness. Biomarkers such as those presented here take advantage

of the water and protein absorption peaks which are prominent spectral features with high detection probability to achieve this. However, it is important to note that these peaks may contain contribution from atmospheric water and a way to minimize this is to measure a background for each sample and to keep the interval between background and sample scan as close as possible to each other. Successful discrimination by this technique could provide a basis for a technology that takes advantage of these prominent peaks for diagnosis of metastasis in primary carcinomas.

The experiments were carried out using different concentrations of cells, which could show the suitability of the spectroscopic technique as a diagnostic technique in the presence of heterogeneity commonly observed in actual cancer tissues. ROC curves show that by carefully choosing an appropriate cutoff for a given spectral property, high sensitivity can be achieved. The choice of the cutoffs was made to optimize sensitivity since an ideal diagnostic procedure should not miss any positive case. It is necessary to note however, that the cutoffs varied across cell lines for a given spectral property and may depend on experimental setup which must be kept uniform for practical purposes. To achieve good cell contact with the diamond ATR element, a screw top knob was used to apply pressure onto samples during measurements. Also, multifractal analysis of spectral images depends on image sampling procedure and requires pixel sizes and other image properties to be kept constant in all cases. This study could be taken further by confirming the results using ex-vivo cells exfoliated from metastatic biopsies from patients and combining all the identified spectral indicators through a machine learning algorithm to give a summative probability of metastasis for a given sample of primary cancer.

## Supporting information

**S1 File. PCA loading vectors of the region 1400 to 1700 cm$^{-1}$.** (A) PC-1 (B) PC-2 loadings for colon cancer cells (C) PC-1 (D) PC-2 loadings for human melanoma cells (E) PC-1 (F) PC-2 loadings for murine melanoma cells (G) PC-3 (H) PC-4 loadings for breast cancer cells. (PDF)

**S2 File. Relative peak and multifractal dimension values for all samples.** (XLSX)

## Acknowledgments

We thank Prof. Elimelech Nesher's laboratory for graciously providing and growing the breast cancer cell lines used in this study.

## Author Contributions

**Conceptualization:** Samuel Onuh Abuh, Ayan Barbora, Refael Minnes.

**Data curation:** Samuel Onuh Abuh.

**Formal analysis:** Samuel Onuh Abuh.

**Funding acquisition:** Refael Minnes.

**Investigation:** Samuel Onuh Abuh.

**Methodology:** Samuel Onuh Abuh, Ayan Barbora, Refael Minnes.

**Project administration:** Refael Minnes.

**Resources:** Refael Minnes.

**Software:** Samuel Onuh Abuh.

**Supervision:** Refael Minnes.

**Validation:** Samuel Onuh Abuh, Ayan Barbora, Refael Minnes.

**Visualization:** Samuel Onuh Abuh, Ayan Barbora.

**Writing – original draft:** Samuel Onuh Abuh.

**Writing – review & editing:** Ayan Barbora, Refael Minnes.

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
