## [Decision Letter · Decision Letter 0]

26 Mar 2024

PONE-D-24-01052Metastasis diagnosis using Attenuated Total Reflection-Fourier Transform Infra-Red (ATR-FTIR) spectroscopyPLOS ONE

Dear Dr. Minnes,

Thank you for submitting your manuscript to PLOS ONE. After careful consideration, we feel that it has merit but does not fully meet PLOS ONE’s publication criteria as it currently stands. Therefore, we invite you to submit a revised version of the manuscript that addresses the points raised during the review process.

We look forward to receiving your revised manuscript.

Kind regards,

Muhammad Muzamil Khan

Academic Editor

PLOS ONE

Journal Requirements:

"The study was partially funded by a research grant from the Administrator General, Israel's Ministry of Justice, application 20220140"

"The study was partially funded by a research grant from the Administrator General, Israel's Ministry of Justice, application 20220140"

"The study was partially funded by a research grant from the Administrator General, Israel's Ministry of Justice, application 20220140."

"The study was partially funded by a research grant from the Administrator General, Israel's Ministry of Justice, application 20220140"

6. We note that your Data Availability Statement is currently as follows: All relevant data are within the manuscript and its Supporting Information files

Reviewers' comments:

Reviewer's Responses to Questions

**Comments to the Author**

1. Is the manuscript technically sound, and do the data support the conclusions?

Reviewer #1: Yes

2. Has the statistical analysis been performed appropriately and rigorously? 

Reviewer #1: No

3. Have the authors made all data underlying the findings in their manuscript fully available?

Reviewer #1: Yes

4. Is the manuscript presented in an intelligible fashion and written in standard English?

Reviewer #1: Yes

5. Review Comments to the Author

Reviewer #1: Dear Authors,

In this study, the use of ATR-FTIR to use as a metstatic prognostic tool in cancer cells was evaluated. You tried to show spectroscopic differences between cancer cells of different metastatic levels. The results support the authors conclusions. I suggest the following minor revisions to increase the clarity of the presentation and highlight the significance of their findings:

1. Firstly, a number of research papers from the same research group was published earlier in 2023 and also in 2017. I would strongly encourage the authors to indicate the novelty of the current study.

2. Figures 4 and 5 both show the mean ratios of the fitted gaussians. Please verify the submission.

3. The authors fail to prove significane in the figures 5 and 6 where they were able to distinguish metastatic cells from non-metastatic.

Sincerely yours

6. PLOS authors have the option to publish the peer review history of their article (what does this mean?). If published, this will include your full peer review and any attached files.

Reviewer #1: No

---

## [Author Response · Author response to Decision Letter 0]

1 May 2024

Response to Reviewer’s Comments on Manuscript PONE-D-24-01052

The Academic Editor

PLOS ONE

Dear Dr. Khan,

Thank you for giving us the opportunity to submit a revision of the draft manuscript ‘Metastasis Diagnosis using Attenuated Total Reflection Fourier Transform Infrared Spectroscopy (PONE-D-24-01052) for publication in Plos One. We appreciate the time and efforts that you and the reviewer dedicated to providing valuable feedback and insightful comments on our manuscript. We have incorporated the suggestions by the reviewer. Furthermore, the multifractal dimensions and water areal ratios for the human melanoma cells were reanalyzed per spectra replicate before averaging, as opposed to previous presentation where the analysis was performed on the average spectra; and linear discriminant analysis has been added at the end of the results section. All changes are highlighted in the manuscript. Please see below a point-by-point response to the reviewer’ comments and the additional concerns you raised. All page numbers refer to the revised manuscript file with tracked changes.

Review Comments to the Author

Reviewer 1

1. Comment from Reviewer 1 pointing out earlier works from our laboratory, and suggesting we indicate the novelty of the current study.

Author Response: The current study extends the works from our group in 2023 and 2017. Spectral markers that were presented in 2017 for melanoma have not been investigated for colon and breast cancer, while the spectral multifractal dimension presented in 2023 has not been investigated for human melanoma and breast cancer cells. This study investigates these markers and additional protein peaks, for each of these cell lines and further evaluated their classification performance using receiver operating characteristic curves and multivariate analysis. Discriminant analysis of the markers has also been added at the end of the results section. Lines 69 to 79 (page 4) of the manuscript have been included to highlight the new task undertaken in the current study. 

2. Comment from Reviewer 1 pointing out a mistake with Figures 4 and 5.

Author Response: Thank you for pointing this out. The Figure referenced 4 was indeed a duplicate of Figure 5. The correct figure has now been included and the mistake fixed.

3. Comment from Reviewer 1 requesting proof of significance.

Author’s Response: The Man-Whitney U test has been used to test the significance of the observed differences in water absorption areal ratios and the mean multifractal dimensions. The respective figures (Fig 5 and 6) have been marked with the results of the significance test.

The Academic Editor’s comments

1. Comment on manuscript style as per Plos One requirements

Author’s Response: The manuscript title page and body has been formatted according to the Plos One style requirements.

2. Comment on code sharing.

Author’s Response: The current manuscript findings are not based on specific codes requiring sharing.

3. Comment on addition of funders’ roles to the financial disclosure.

Author’s Response: The updated financial disclosure is as follows:

‘The study was partially funded by a research grant from the Administrator General, Israel’s Ministry of Justice, application number 20220140. The funders had no role in study design, data collection and analysis, decision to publish, or preparation of the manuscript.’

4. Comment on funding statement.

Author’s Response: The updated Funding Statement is as follows:

‘The study was partially funded by a research grant from the Administrator General, Israel’s Ministry of Justice, application number 20220140. There was no additional external funding received for this study.’

5. Comment on funding information in the Acknowledgement Section

Author’s Response: The Acknowledgement Section has been updated and the funding information in the section has been removed.

6. Comment on Data Availability Statement

Author’s Response: We confirm that all relevant data are within the manuscript and its supporting information files.

7. Comment on the Reference List

Author’s Response: The reference list has been reviewed and the following updates effected according to the Vancouver style. Reference number 11 was changed to doi.org/10.1039/C7AN01871A. The initial reference was an associated correction article (located at https://doi.org/10.1039/C8AN90029A) for the now presented reference. References 4 and 5 were changed to more current and relevant ones, page number included in reference 14 and additional references including 42 to 44 were added to further buttress the discussion section.

---

## [Editor Report · Decision Letter 1]

7 May 2024

Metastasis diagnosis using Attenuated Total Reflection-Fourier Transform Infra-Red (ATR-FTIR) spectroscopy

PONE-D-24-01052R1

Dear Dr. Refael Minnes

We’re pleased to inform you that your manuscript has been judged scientifically suitable for publication and will be formally accepted for publication once it meets all outstanding technical requirements.

Kind regards,

Muhammad Muzamil Khan

Academic Editor

PLOS ONE
---

## [Editor Report · Acceptance letter]

10 May 2024

PONE-D-24-01052R1 

PLOS ONE

Dear Dr. Minnes, 

I'm pleased to inform you that your manuscript has been deemed suitable for publication in PLOS ONE. Congratulations! Your manuscript is now being handed over to our production team.

Kind regards, 

on behalf of

Dr. Muhammad Muzamil Khan 

Academic Editor

PLOS ONE